# Peritraumatic Distress during the COVID-19 Pandemic in Seoul, South Korea

**DOI:** 10.3390/ijerph18094689

**Published:** 2021-04-28

**Authors:** Hyejung Yoon, Myoungsoon You, Changwoo Shon

**Affiliations:** 1The Seoul Institute, 57 Nambusunhwan-ro, 340-gil, Seocho-gu, Seoul 06756, Korea; yoonhj838@si.re.kr; 2Department of Public Health Science, Graduate School of Public Health, Seoul National University, 1 Gwanak-ro, Gwanak-gu, Seoul 08826, Korea; msyou@snu.ac.kr

**Keywords:** COVID-19, peritraumatic distress, peritraumatic distress inventory, social distancing, infectious disease

## Abstract

The coronavirus disease (COVID-19) pandemic is traumatic and causes a substantial psychological burden on the general public. The aim of the present study is to examine the severity and prevalence of peritraumatic distress among the citizens of Seoul, which conducted preemptive and aggressive social distancing policy before the central government during the early stage of COVID-19. Furthermore, this study aims to explore the associated risk factors for peritraumatic distress, including risk perception, fear, and COVID-19-related experiences. We conducted an online survey to 813 participants at the end of the first wave of COVID-19 in South Korea. Peritraumatic distress inventory (PDI) was used to measure the level of pandemic-related distress. One-third of participants were at risk for the development of clinically elevated peritraumatic distress. The perception of risk, fear of COVID-19, and stigma were significantly associated with elevated levels of distress. Individuals who had poor health, or who spent more than 1 h per day using the media, also expressed a higher level of distress. Moreover, the level of disruption of daily life and financial difficulties due to the COVID-19 pandemic is significantly associated with a higher level of peritraumatic distress. The results of this study highlight the urgent need to develop evidence-based and tailored public mental health interventions, along with various measures to help recovery to daily life.

## 1. Introduction

Since the first confirmed case emerged in Wuhan, China, in December 2019, the novel coronavirus disease (COVID-19) has spread around almost all of the world at an alarmingly fast rate. It was declared a global pandemic 72 days after the first case by the World Health Organization on 11 March 2020 [1], and it is the second pandemic of the 21st century officially announced by WHO, following the H1N1 virus pandemic in 2009–2010 [2]. Countries have been afflicted with COVID-19 for over one year and now undergoing a third wave of COVID-19. As of 3 February 2021, there have been more than 100 million confirmed cases around the world [3], and 79,311 confirmed cases, including 1441 deaths, in South Korea [4].

The COVID-19 pandemic has been causing a substantial psychological burden and threatening the mental health of individuals [5,6]. The novel virus has invaded into the daily social life of individuals and compelled massive changes to normal daily life such as financial difficulties; loss of social welfare services, including childcare and education, due to the closure of public facilities; and changes in the way of working and employment [7]. Stringent and high-intensity public health measures, which include social distancing and lockdown, have been implemented worldwide to control the spread of COVID-19. The previous literature indicates that, along with the fear of infection, these unprecedented changes to daily life during the outbreak of infectious diseases can negatively affect individuals’ mental health and cause psychological problems (e.g., distress, anxiety, depression, posttraumatic stress, etc.) [8]. Frontline healthcare workers, people who are quarantined, and survivors are known to be a high-risk group in terms of mental health both during and after the occurrence of novel infectious virus disease, such as severe acute respiratory syndrome (SARS) [9,10,11,12,13,14,15,16] and Middle East respiratory syndrome (MERS) [17,18].

Recent research studies into the adverse psychological consequences for the general population during the early stage of COVID-19 have also shown that anxiety, stress, and depression are common mental health problems [6]. Nationwide large-scale surveys in China have found that a significant number of the general population have experienced psychological distress, including anxiety, depression, and stress [5,19,20,21]. Psychiatric symptoms that include psychological reactions similar to those in China have also been reported in many other countries, including Italy [22], Spain [23], the USA [24,25], Australia [26], and Israel [27]. 

Recently, trauma-related mental health problems due to COVID-19 such as post-trauma stress disorder and traumatic stress also has been examined. The COVID-19 pandemic creates substantial traumatogenic events for everyone, from confirmed patients and healthcare workers to the general public [28,29,30]. Nevertheless, trauma-related reactions and posttraumatic stress disorder (PTSD) due to the novel coronavirus have been addressed relatively less compared to the general psychological problems such as depression and anxiety [30]. Considering the facts that the psychological impact of COVID-19 on individuals could vary in degree and in type [31] and that the pandemic is likely to be a traumatic stressor that is still in progress [28], it is critical to examine trauma-related reactions due to COVID-19, along with psychological reactions such as anxiety, depression. 

Peritraumatic distress is the emotional and physiological distressed responses during and/or immediately after a traumatic event [28,32]. Although peritraumatic distress is not a clinically diagnosed psychiatric disease, it has been demonstrated to be associated with a higher risk of developing PTSD [33] and increased PTSD severity [34]. Considering the fact that the COVID-19 pandemic is still in progress, it is likely that general people may experience traumatic distress in their daily life.

There is still an urgent need for more research that explores the traumatic impact of COVID-19 [30] as well as mental health problems associated with the COVID-19 outbreak in other affected countries [6,35]. Moreover, few research studies have examined the trauma-related psychological problems of the general population of South Korea during the COVID-19 pandemic, which is one of the countries that COVID-19 struck quickly. Characteristics of the megacities such as Seoul—which have, for example, a high population density, a large floating population, and an active and intensely social, economic, and cultural exchange globally—could be a trigger for the rapid transmission of the virus [36], leading to the elevated mental health problem issues. Since the largest mass infection cases emerged in the city of Daegu in February 2020, which was the epicenter of the COVID-19 outbreak in South Korea and the first large outbreak case of the virus outside of China [37], new cluster infection cases have appeared continuously in Seoul, such as the case of a large company’s call center in March, a case of bars and clubs in a multicultural district in May, and most recently, a case of the religious institutions in August. While experiencing the MERS outbreak in 2015, the city of Seoul has strived constantly for the establishment of a system that operates in public health crises such as new infectious diseases [38]. The Seoul Metropolitan Government has been taking a proactive and stringent COVID-19 prevention and surveillance approach [38]. However, despite its endeavor, Seoul also has faced the third wave of COVID-19 along with other megacities. Seoul became the city with the most COVID-19 confirmed cases in South Korea, and as of 4 February 2021, there have been 24,583 confirmed cases in Seoul [4].

The present study aimed to explore the severity and prevalence of peritraumatic distress among the general population in Seoul, South Korea, during the COVID-19 pandemic. We also examined the associated risk factors, including sociodemographic characteristics, health status, risk perception and fear of COVID-19, and other COVID-19-related factors. We expect to provide the grounds for developing evidence-based psychological interventions and strategies for the general population by identifying vulnerable groups and the risk factors that are associated with peritraumatic distress during the COVID-19 pandemic. 

## 2. Materials and Methods

### 2.1. Design and Participants

The survey was conducted by the Seoul Institute and Seoul National University via an online platform. Respondents were recruited from an online panel of a professional public opinion company called Hankook Research using proportionate quota sampling that was based on age, sex, and the five regions of Seoul, and the inclusion criteria were being aged 18 years or older and living in Seoul. Potential respondents in the panel were invited via e-mail and text messages that contained a designated link with brief information about the current study. Respondents were provided detailed information about the study when they visited the online survey site. Only those who check the consent to voluntarily participated in the survey and to use their response data only for statistical and research purposes could participate in the survey. Data were collected just before the transition from intensive social distancing to social distancing in daily life (6 May) from 28 April 2021 to 1 May 2021. In total, 1007 people participated in the survey. After removing participants with incomplete questionnaires, 813 adult citizens from all 25 districts of Seoul were included in the analysis. The survey process and informed consent to participate voluntarily were approved by the institutional review board of Seoul Medical Center (no. 2020-04-005). The study was conducted in accordance with the principles of the Declaration of Helsinki.

### 2.2. Questionnaire

#### 2.2.1. Sociodemographic Data

In addition to sex, age, and region, which were used for the quota sampling, sociodemographic data were collected that involved educational attainment, marital status, household composition—that is, the presence of cohabitants and presence of children aged 18 years and under—monthly household income, and employment status. The monthly household income was converted into the monthly household equivalent income by dividing the median income for each range by the number of household members, and we reclassified it into four categories (the values ranged from 1 (less than USD 1714) to 4 (more than USD 5143)). We also collected information about the general health and mental health status of the participants. Participants were asked whether they had any chronic diseases that were diagnosed by a physician or had received treatment for chronic disease in the last year, and they completed a single-item measure that assessed self-rated health status on a five-point Likert scale that ranged from “bad” to “excellent.” Responses to the self-rated health status item were recoded as a dummy variable: 0 for “bad health status” and 1 for “good health status.” Mental health status was evaluated by asking participants whether they had experienced a psychiatric visit or psychological counseling for mental health problems in the past year.

#### 2.2.2. Peritraumatic Distress

Peritraumatic distress related to the COVID-19 pandemic was assessed using the 13-item peritraumatic distress inventory (PDI) [33]. PDI measures the level of emotional and physiological distress experienced by an individual during and/or after a traumatic event [32,33], such as COVID-19. The items cover negative emotions (e.g., “I felt helpless to do more,” “I felt sadness and grief”) and perceived threat to life and bodily arousal (e.g., “I felt afraid for my safety,” “I thought I might die”) [32,33]. PDI has demonstrated good internal consistency with test–retest reliability and good convergent and divergent validity [33]. The original PDI items were translated into Korean using the back-translation procedure by public health experts who are fluent in both languages to ensure consistency of meaning. In this study, the PDI has excellent internal consistency, with a Cronbach’s alpha of 0.91.

To capture distress due to the COVID-19 pandemic, we modified the original PDI instruction (i.e., “A lot has happened during the 100 days since the outbreak of the COVID-19 epidemic in our society. To what extent have you experienced the following items during the COVID-19 epidemic?”) [28,29]. The response options ranged from 0 (not at all true) to 4 (extremely true), and the ratings were summed to calculate the total score of peritraumatic distress. Higher total scores indicate a highly distressed status. Prior researches suggested that a cutoff score of 23 is a useful threshold to predict the onset of posttraumatic distress disorder (PTSD) [32,39]. The aim and scope of this study do not include the clinical confirmation of the definite diagnosis of trauma-related stress disorders. Therefore, we only present how many people were at risk for the development of clinically elevated peritraumatic distress based on the cutoff score of 23 in the results section. Total score and subtotal score for each subfactor of PDI were used to investigate the associated risk factors in regression analysis. 

#### 2.2.3. Risk Perception and Fear of COVID-19

Following Lee and You [40], the perceived risk of COVID-19 was assessed using two questions (i.e., perceived susceptibility and perceived severity). Participants rated the possibility and severity of being infected with the novel COVID-19 virus on a five-point Likert scale that ranged from 1 (not at all) to 5 (extremely). 

We measured two aspects of the extent of fear of COVID-19: fear of infection and fear of social stigma. Fear of infection was evaluated using the following two questions that were responded to on a five-point Likert scale: (1) “I am afraid of being a confirmed COVID-19 patient” and (2) “I am afraid that there will be more confirmed cases in my neighborhood.” Fear of stigma was also assessed using two questions: (1) “If I become a confirmed COVID-19 patient, I am afraid that I will be criticized based on this fact” and (2) “If there are confirmed patients in my neighborhood, I am afraid that my neighborhood will be criticized based on this fact.”

#### 2.2.4. COVID-19-Related Experience, Social Support, and Media Use

We asked the participants about experiences related to COVID-19 by questioning whether they or their close friends had ever contracted it and/or been quarantined (0 = no, 1 = yes). They also rated whether the COVID-19 pandemic had negatively influenced their financial situation and whether their income had decreased due to COVID-19 on a five-point Likert scale. We also asked them to what extent their daily life was disrupted by the COVID-19 pandemic using a visual slider scale, which ranged from 0 (complete disruption of daily life) on the left-hand side to 100 (no disruption) on the right-hand side. The responses were reverse coded.

The participants’ perceived social support was assessed by asking whether they have anyone, except for family members, who they could urgently ask for help if they were quarantined (0 = no, 1 = yes) [40]. Regarding media use, we asked the participants how much time they spent using media for news or information about COVID-19. The response options were “within 30 min,” “within 1 h,” “1 to 2 h,” “2 to 4 h,” and “more than 4 h.” We recategorized the responses into “within 1 h” and “more than 1 h.”

### 2.3. Statistical Analysis

The data were analyzed using the SAS 9.4 software (SAS Institute Inc., Cary, NC, USA) for all of the statistical analyses. First, descriptive statistics were used to present the sociodemographic characteristics and COVID-19-related status of the participants, as well as the mean scores of PDI. Second, multivariate linear regression was conducted to investigate the association between the COVID-19-related variables and the level of peritraumatic distress after controlling the sociodemographic variables. There was no multicollinearity among the independent variables. *p* < 0.05 was considered statistically significant in all of the analyses.

## 3. Results

### 3.1. Sample Characteristics

The sociodemographic characteristics of the study sample are presented in Table 1. The participants’ sex and age groups were evenly distributed, and their ages ranged from 18 to 81 years, with a mean age of 46.0 ± 14.9 years old. The educational background of the participants was relatively high, and more than 70% were college educated or above. Regarding the household composition, 11.9% lived in “adult-only” households, and 22.4% had children aged 18 years old and under. In terms of monthly household income, USD 1714–3427 was the most common range, and USD 3428–5142 was the least common range. Of the participants, 39.6% were full-time employees, and 37.0% were unemployed. Almost half of the sample (47.4%) regarded their health status as bad, while 10.1% and 17.6% of the sample reported that they had more than two chronic diseases or mental health problems, respectively. The level of psychological distress was significantly different according to the sociodemographic characteristics of sex, age, and health-related variables.

### 3.2. Peritraumatic Distress

The mean scores and standard deviations (*SD*) of the participants’ answers for each item of the PDI are shown in Table 2. The average total score of PDI was 19.75 (*SD* = 8.82, ranging from 0 to 52). Among the 13 items, those related to concerns about safety (i.e., Item 7 “I felt worried about the safety of others” (mean (*M*) = 2.65, *SD* = 0.82) and Item 4 “I felt afraid for my safety” (*M* = 2.18, *SD* = 1.03)) and one item related to helplessness (Item 1 “I felt helpless to do more” (*M* = 2.03, *SD* = 1.03)) had scores higher than 2. On the other hand, items related to the physical reaction (i.e., Item 9 “I had difficulty controlling my bowel and bladder” (*M* = 0.76, *SD* = 0.85), Item 11 “I had physical reactions such as sweating, shaking, and pounding heart” (*M* = 0.83, *SD* = 0.92), and Item 12 “I felt I might pass out” (*M* = 0.83, *SD* = 0.94)) were rated lower than 1.

When using the PDI cutoff score of 23 [32,39], 34.4% of the participants (*n* = 280) were at risk for the development of clinically elevated peritraumatic distress.

### 3.3. Risk Perception, Fear, and COVID-19-Related Experience

The participants’ perceived susceptibility was higher than “low” (*M* = 2.70, *SD* = 0.74, range = 1–5, where 1 = not at all and 5 = extremely). Over half of them reported that the possibility of infection was neither high nor low (53.6%), and only 10.1% reported it to be “high” or “very high.” The perceived severity of being infected with SARS-CoV-2 was close to “high,” with an average score of 3.87 (*SD* = 0.82). The majority of participants rated the severity as “high” (54.0%), which was followed by “very high” (20.2%), and a very small proportion reported that it was “not at all” serious. 

The overall level of fear of contracting COVID-19 was higher than “moderate” (*M* = 3.56, *SD* = 0.89). The participants expressed a similar level of concern for themselves and their neighbors (*M* = 3.58, *SD* = 1.00 vs. *M* = 3.55, *SD* = 0.99). They were also worried that if they or their neighbors contracted COVID-19, they could be socially criticized (*M* = 3.27, *SD* = 0.95). However, unlike the fear of the infection itself, the fear of the stigma that they would experience if they became infected with SARS-CoV-2 was higher than the fear of the stigma that their neighbor would receive (*M* = 3.57, *SD* = 1.10 vs. *M* = 2.97, *SD* = 1.10).

Regarding the extent of the disruption of daily life since the outbreak of the COVID-19 pandemic, participants reported that their daily routine changed moderately (*M* = 46.61, *SD* = 23.28, range = 0–100, where 0 = no disruption and 100 = completely disrupted). The participants’ average score of financial difficulties during the COVID-19 pandemic was 3.51 (*SD* = 1.00). 

The percentage of participants who were confirmed patients of COVID-19, and/or had isolated voluntarily or compulsorily, and/or whose family or friends were confirmed patients or had isolated was 19.9%. Most of them (91.8%) reported that there was more than one person, except for family members, whom they could urgently ask for help in case of quarantine due to COVID-19. Three-quarters of the sample used media for COVID-19-related information and the time spent was “within 1 h” on average per day.

Bivariate correlation analysis showed significantly moderate positive associations between variables (*r* from 0.09 to 0.52, *p* < 0.05). Detailed results of correlations, including means and standard deviations for risk perception, fear, COVID-19-related experience, and distress are shown in Table 3. 

### 3.4. Results of the Multivariate Regression Analysis

The multivariate regression analysis showed an adequate adjustment in general, and it explained 42.1% of the variance, *F* (29, 783) = 19.6, adjusted *R*^2^ = 0.42, *p* < 0.0001. The results of the associations between risk perception, fear of infection, fear of social stigma, the experience of COVID-19, disruption of daily life, financial difficulties, social support, media consumption, and peritraumatic distress, when controlling all sociodemographic characteristics, are displayed in Table 4. 

As shown in Table 4, the individual perception of the susceptibility and severity of COVID-19 were significantly associated with the level of peritraumatic distress, that is, the higher the perceived susceptibility and severity were, the more individuals felt distressed. Fear of infection with the COVID-19 virus and fear of social stigma was associated with higher trauma-related distress, as were perceived disruption of daily life and financial difficulties. Having experience of COVID-19, such as being a confirmed case and being quarantined, is also related to the increased level of distress. Furthermore, media use of more than 1 h per day for COVID-19-related information was associated with an increased level of distress. Among the sociodemographic variables, only factors related to the health status—the number of chronic diseases, subjective health status, and history of mental health problems—were significantly associated with peritraumatic distress. Individuals with more than two chronic diseases and a history of mental health treatment had a higher level of distress, whereas individuals with good subjective health had a lower level of distress.

Among psychological variables, having higher perceived severity and fear of COVID-19 increased respondents’ peritraumatic distress through both negative emotions and life threat and physical arousal. COVID-19-related experience showed a different influence on subdomains of PDI. Perceived level of disruption of daily life and financial problems due to COVID-19 were associated with a higher level of negative emotions, whereas the experience of COVID-19 and media consumption were associated with both subdomains of PDI. Among the sociodemographic variables, the educational attainment of respondents and objective health status were associated with the negative emotions, and subjective health status and mental health history were associated with the domain of life threat and physical arousal. 

## 4. Discussion

This study aimed to investigate the peritraumatic reaction of the general population of Seoul, South Korea, and the associated factors during the COVID-19 pandemic. We applied the PDI to capture how distressed the people due to the traumatic-related features of the COVID-19 pandemic. Considering the cutoff score of 23 of the PDI [32,33], one-third of the participants in this study were at risk of developing PTSD in this study. Peritraumatic distress has been demonstrated to be associated with a higher risk of developing PTSD [33,41], and negative emotions as a component of peritraumatic distress were the second-best predictor of PTSD [42]. In a study that investigated PTSD risk among the general population in the early stage of the COVID-19 pandemic, 29.5% of participants in Italy [43] and 17.7% in Ireland [44] were at high risk for developing PTSD. 

Based on the average total score of the PDI, the participants of the study were more distressed due to the COVID-19 pandemic, compared to previous studies conducted in other countries (i.e., 8.9 ± 7.3 in Japan [45], 9.5 ± 6.5 [27] and 11.4 ± 7.9 [46] in Israel, and 18.9 ± 5.4 for men and 22.5 ± 7.1 for women in Greece [47]). Considering the differences in the COVID-19 pandemic situation in each country at the time of data collection and the differences in the capacity, including that of the healthcare system, to respond to new infectious diseases by country, it is understandable that the level of peritraumatic distress differed by the study. However, the survey was conducted almost 3 months after the first COVID-19 case in South Korea (20 January 2020). It was at this time that the spread of the infection seemed to be controlled to some extent in Korea as the new daily cases of COVID-19 had decreased; as of April 28, the average number of newly confirmed cases in a week was less than 10.0 per day. As a result, alleviating social distancing measures, namely, “distancing in daily life,” was actively discussed at that time. The result of the study shows that the individuals’ negative psychological reactions and life threat experience related to COVID-19 can persist even after the peak of infection due to COVID-19.

Our results indicate that fear of contracting the disease and fear of stigma, along with the perception of risk related to COVID-19, were risk factors for individuals’ elevated peritraumatic response during the COVID-19 pandemic. Similar to our study, in Australia [26], individuals’ concerns or worries about COVID-19 are associated with higher levels of depressive symptoms, anxiety symptoms, and stress. The manifestation of negative responses, such as stress, anxiety, and/or fear, is a natural reaction as a human being in a highly uncertain epidemic situation, such as the COVID-19 pandemic [48]. These negative psychological responses help motivate people to take an active and voluntary role in preventive behaviors during an epidemic [40,49,50,51]. However, overestimated fear and risk perception can be harmful not only at the individual level but also at the societal level [48,52]. In the situation of a highly uncertain epidemic, it is likely that people’s amplified and uncontrolled fear could lead to the stigmatization and alienation of those who are confirmed cases, quarantined, or in close contact with confirmed patients [48]. 

In relation to the COVID-19 pandemic, Korean society has experienced several serious crises of social dichotomy according to religion, sexual identity, age group, or political orientation [53]. Fear for COVID-19 and fatigue due to prolonged and strengthened quarantine measures led to a ruthless social criticism of specific infectious groups. Since quarantine was prioritized over individual freedoms and human rights [53,54], people who related to widespread infection were stigmatized as a public enemy [53]. Furthermore, the spread of misinformation through the media and social networks, and victim blaming deepened the social division. Social exclusion, prejudice, suspicion, and discrimination from neighbors and society are as great stressors as fear of the COVID-19 infection for the general population [8,30] in the pandemic situation. It is necessary to block the spread of false information thoroughly and increase solidarity and community efficacy to prevent irrational social stigma. 

Providing accurate and sufficient information promptly is a crucial factor for lowering the public’s fear and risk perception. For unified and informative messaging based on facts, the Central Disaster Safety and Countermeasure Headquarters (CDSCHQ), along with the Central Disease Control Headquarters in Korea, has disclosed COVID-19-related information every day since the first case appeared. Most of the information is about the occurrence of newly confirmed cases and the results of the epidemiological investigation. Based on the CDSCHQ reports, the media continuously publish related content with experts’ opinions. However, considering the media’s tendency to use negative and sensitive terms and provocative statements frequently when reporting infectious diseases [55], repetitive and excessive exposure to the media for information about COVID-19 could have a negative impact on mental health by amplifying the fear and making people feel anxious [19,52].

Until the recent development of the COVID-19 vaccines, social distancing was the only way to prevent the spread of infections in addition to personal preventive behaviors such as wearing face masks and handwashing. However, social distancing might be attributed to the elevated risk of mental health problems such as social withdrawal and family conflict [8]. It has also resulted in enormous costs to individuals and society as a whole by suppressing social, cultural, and economic activities, and individuals are exposed to the traumatic events due to financial difficulties such as the closing of the business and unemployment [35]. In this study, it is also confirmed that the financial problems related to the COVID-19 pandemic are associated with elevated peritraumatic distress. In particular, when we examined the level of financial difficulties by employment type, the self-employed and part-time employees experienced a lot of financial problems during the COVID-19 pandemic.

To mitigate the COVID-19-related economic damage to individuals, households, and businesses, various financial support plans have been devised and implemented in many countries. In the case of South Korea, as the aggressive social distancing comparable to the shutdown has been carried out for a long time, the self-employed are threatened with livelihood due to business suspension. The emergency disaster relief fund was provided to everyone regardless of income and property in May 2020, and lots of financial support measures such as low-interest loans for small business owners and employment maintenance support to prevent unemployment are being implemented. Financial aid through cash payments is helpful to individuals and businesses that have been severely affected by COVID-19. However, considering the unknown timeline of the COVID-19 pandemic and the effect of vaccination, it is necessary to seek and implement additional support measures to maintain economic activities along with financial supports.

Among the sociodemographic variables, health-related factors were associated with psychological distress. The participants who had more than two chronic diseases or a prior history of psychological treatment were more psychologically distressed than those who had no such health problems. These findings are in line with the results of previous studies [21,22,26]. Individuals with multiple chronic conditions are likely to have greater health needs, and studies indicate that multimorbid people had higher rates of healthcare utilization in terms of inpatient, outpatient, and emergency room visits [56,57,58]. In addition, the rates of consultations with doctors are very high in Korea. Considering these conditions, it is possible that those who have a poor health status might be more distressed as they cannot obtain the necessary healthcare services, or they have to visit hospitals despite the possibility of infection.

To the best of our knowledge, this paper presents the first insight into the traumatic reaction related to the COVID-19 pandemic in the general population of Seoul, South Korea. According to Brooks et al., [8], quarantine, which is a countermeasure for lessening the spread of the infectious virus, could have negative psychological consequences, and the effect could persist for both individuals and society. Based on their comprehensive literature review of the adverse effects of quarantine on mental health during past epidemics, such as SARS, H1N1, and MERS, they warned that the psychological costs of imposed mass quarantine could offset or exceed the expected effects. In this study, we have also confirmed the negative effect of the disruption of daily life on individuals’ mental health during the mass quarantine. 

There are some limitations to consider when interpreting the results of this study. First, since this was a cross-sectional study that captured the participants’ perception only at the time that the survey was performed, we cannot establish causal relationships among the variables. Second, since we assessed only the relative symptoms of peritraumatic distress that were associated with the COVID-19 pandemic, the findings should not be interpreted as indicative of a traumatic distress disorder. Furthermore, the number of participants aged 65 years old and over was less than those of other age groups. Since the survey was conducted online, access to the survey would have been more limited by elderly people who are not familiar with the use of digital devices. Future studies should investigate the longitudinal effect of the COVID-19 pandemic on the mental health of the general public in addition to its impact on healthcare professionals, epidemiological investigators, and public officials.

Despite these limitations, the findings of the current study improve our understanding of the traumatic effect of the COVID-19 pandemic on the general public. The identified risk factors were in line with those of previous studies conducted in other countries. Currently, COVID-19 is rapidly spreading again in many countries, including Korea. The re-emergence of COVID-19 along with a prolonged mass quarantine could be a psychological shock to people who expected to recover their normal daily life. Therefore, it is essential to develop evidence-based and targeted public mental health interventions to preserve the psychological health of the public. Furthermore, individuals, groups, or regions with multiple risk factors for peritraumatic distress identified in the present study should be identified and connected to appropriate community services such as psychological counseling.

## 5. Conclusions

In the era of the COVID-19 pandemic, problems related to psychosocial well-being such as distress, anxiety, and depression have become a universal phenomenon that anyone can experience. This study examined the peritraumatic distress of the citizens of Seoul, which conducted preemptive and extensive social distancing campaigns before those of the central government during the early stage of COVID-19. We confirmed that the traumatic distress appears differently depending on the social and psychological characteristics related to the COVID-19 of individuals. Differentiated and multilevel approaches are needed for each group, and above all, mental health interventions for high-risk groups must be urgently implemented through connection with local medical institutions. Since the whole world is experiencing the reemergence of COVID-19, along with unexpectedly prolonged social distancing, it is essential to develop evidence-based and targeted public mental health interventions, along with various measures to help recovery daily life. 

## Figures and Tables

**Table 1 ijerph-18-04689-t001:** Descriptive statistics and group differences in the scores on the PDI (*n* = 813).

Characteristics	*n*	*%*	M (SD)	t/F
Sex				
Men	377	46.4	18.97 (9.33)	−2.32 *
Women	436	53.6	20.42 (8.31)	
Age Groups	*M* = 46.0	*SD* = 14.9		
18–29	152	18.7	18.78 (8.80)	2.88 *
30–39	149	18.3	20.61 (9.68)	
40–49	152	18.7	18.11 (8.87)	
50–59	146	18.0	20.12 (8.71)	
Over 60’s	214	26.3	20.73 (8.08)	
Highest level of educational attainment				
Under high school	215	26.5	19.25 (8.72)	0.69
College	486	59.8	19.81 (8.99)	
Graduate School and above	112	13.8	20.44 (8.30)	
Marital status				
Single ^a^	332	40.8	19.37 (8.98)	−1.00
Married	481	59.2	20.00 (8.71)	
Presence of Cohabitants				
No	97	11.9	19.29 (9.39)	−0.54
Yes	716	88.1	19.81 (8.75)	
Presence of children aged ≤ 18				
No	631	77.6	19.75 (8.89)	0.05
Yes	182	22.4	19.72 (8.62)	
Monthly household equivalence income				
Under USD 1714	161	19.8	19.64 (9.15)	0.68
USD 1714–3427	407	50.1	19.97 (8.72)	
USD 3428–5142	106	13.0	18.66 (8.88)	
USD ≥5143	139	17.1	20.04 (8.73)	
Employment status				
Full-time employees	322	39.6	19.18 (9.33)	1.28
Part-time employees	104	12.8	19.15 (8.05)	
Self-employed	86	10.6	20.60 (7.90)	
Unemployed ^b^	301	37.0	20.31 (8.76)	
Health status				
Bad	385	47.4	21.63 (8.94)	5.90 ***
Good	428	52.6	18.05 (8.37)	
Number of chronic diseases				
0	549	67.5	19.20 (8.78)	7.56 ***
1	182	22.4	19.81 (9.02)	
≥2	82	10.1	23.23 (7.98)	
Mental health problem history ^c^				
No	670	82.4	19.37 (8.48)	−2.38 *
Yes	143	17.6	21.52 (10.12)	

USD: US dollars (USD 1 = Korean won (KRW) 1166.72 based on the basic exchange rate in 2019); *M*: mean; *SD*; standard deviation; ^a^: including “never married”, “divorced or separated”, and “widowed”; ^b^: including “housewives,” “students,” “unpaid family worker,” and “retired”; ^c^: experience of mental health treatment of psychological counseling in the past year; * *p* < 0.05, ** *p* < 0.01, *** *p* < 0.001.

**Table 2 ijerph-18-04689-t002:** Peritraumatic distress (*n* = 813).

Items	*M* ± *SD*
Negative emotions	11.42 ± 5.38
I felt helpless to do more.	2.03 ± 1.03
I felt sadness and grief.	1.97 ± 1.00
I felt frustrated or angry I could not do more.	1.74 ± 1.08
I felt guilt that more was not done.	1.53 ± 0.97
I felt ashamed of my emotional reactions.	1.17 ± 0.91
I had the feeling I was about to lose control of my emotions.	1.15 ± 0.97
I was horrified by what happened.	1.83 ± 1.04
Life threat and physical arousal	8.32 ± 4.02
I felt afraid for my safety.	2.18 ± 1.03
I felt worried about the safety of others.	2.65 ± 0.82
I had difficulty controlling my bowel and bladder.	0.76 ± 0.85
I had physical reactions like sweating, shaking, and pounding heart.	0.83 ± 0.92
I felt I might pass out.	0.83 ± 0.94
I thought I might die.	1.08 ± 1.00
Total	19.75 ± 8.82
<23	65.6%
≥23	34.4%

**Table 3 ijerph-18-04689-t003:** Correlations between risk perception, fear, COVID-19-related experience, and PDI.

	1	2	3	4	5	6	7	8	9	10
1. Perceived susceptibility	-									
2. Perceived severity	0.22 ***	-								
3. Fear of infection	0.29 ***	0.39 ***	-							
4. Fear of social stigma	0.19***	0.24 ***	0.61 ***	-						
5. Disruption of daily life	0.13 ***	0.14 ***	0.21 ***	0.18 ***	-					
6. finalcial problem	0.12 ***	0.16 ***	0.16 ***	0.19 ***	0.35 ***	-				
7. Experience of COVID-19 ^a^	0.09 *	−0.09 *	−0.01	0.02	0.01	−0.01	-			
8. Social support ^a^	0.03	−0.01	−0.07	−0.09 **	−0.03	−0.04	-	-		
9. Media consumption ^a^	0.07 *	0.05	0.09 *	0.11 **	0.03	0.05	-	-	-	
10. *PDI*	0.27 ***	0.32 ***	0.52 ***	0.47 ***	0.24 ***	0.26 ***	0.13 ***	−0.09 *	0.17 ***	-
*M* (*SD*)	2.70 (0.74)	3.87 (0.80)	3.56 (0.89)	3.27 (0.95)	46.61 (23.28)	3.51 (1.00)	0.20 (0.40)	0.92 (0.28)	0.25 (0.43)	19.75 (8.82)

Note. *PDI* = peritraumatic distress inventory; *M*: mean; *SD*; standard deviation; ^a^ point-biserial coefficient; * *p* < 0.05, ** *p* < 0.01, *** *p* < 0.001.

**Table 4 ijerph-18-04689-t004:** Results of multivariate regression analysis (*n* = 813).

Variables	Negative Emotions			Life Threat and Physical Arousal			Peritraumatic Distress		
	Estimate	S.E.	Pr > |t|	Estimate	S.E.	Pr > |t|	Estimate	S.E.	Pr > |t|
Intercept	−5.698	1.355	< 0.0001	−2.490	0.993	0.012	−8.189	2.136	0.000
Gender (Male, reference)									
Female	−0.044	0.326	0.894	0.172	0.239	0.472	0.128	0.513	0.803
Age group (19–29, reference)									
30–39	0.088	0.572	0.879	−0.122	0.419	0.771	−0.035	0.902	0.969
40–49	−0.322	0.621	0.604	−0.589	0.455	0.196	−0.912	0.979	0.352
50–59	−0.284	0.638	0.656	−0.278	0.467	0.552	−0.562	1.005	0.576
Over 60’s	−0.433	0.660	0.512	−0.768	0.484	0.113	−1.201	1.041	0.249
Educational attainment (Under high-school graduate, reference)									
College graduate	1.236	0.392	0.002	−0.084	0.287	0.770	1.152	0.618	0.063
Graduate school or above	1.126	0.549	0.041	−0.062	0.403	0.877	1.064	0.866	0.220
Marital status (Unmarried, reference)									
Married	−0.242	0.483	0.616	0.516	0.354	0.145	0.274	0.761	0.719
Presence of cohabitant (No, reference)									
Yes	0.574	0.525	0.274	−0.543	0.385	0.159	0.032	0.827	0.969
Presence of children aged ≤ 18 (No, reference)									
Yes	−0.398	0.491	0.418	0.081	0.360	0.821	−0.316	0.774	0.683
Monthly household equivalence income (Under USD 1714, reference)									
USD 1714–3427	0.618	0.424	0.146	0.267	0.311	0.121	0.885	0.669	0.186
USD 3428–5142	−0.351	0.575	0.542	−0.359	0.421	0.739	−0.710	0.906	0.434
USD ≥5143	1.131	0.542	0.037	−0.036	0.397	0.693	1.096	0.855	0.200
Employment status (Full-time employees, reference)									
Part-time employees	−0.444	0.513	0.386	−0.583	0.394	0.121	−1.027	0.808	0.204
Self-employed	0.092	0.563	0.870	0.138	0.413	0.739	0.230	0.888	0.796
Unemployed	0.793	0.417	0.057	−0.121	0.305	0.693	0.673	0.657	0.306
Subjective health status (Bad, reference)									
Good	−0.585	0.327	0.074	−0.967	0.240	<0.0001	−1.552	0.516	0.003
No. of chronic diseases (0, reference)									
1	0.056	0.420	0.894	−0.021	0.308	0.946	0.035	0.662	0.958
≥2	1.308	0.587	0.026	0.658	0.430	0.126	1.967	0.925	0.034
Mental health history (No, reference)									
Yes	0.810	0.411	0.050	1.059	0.302	0.001	1.869	0.649	0.004
Perceived susceptibility	0.427	0.220	0.053	0.497	0.162	0.002	0.924	0.348	0.008
Perceived severity	0.500	0.209	0.017	0.529	0.153	0.001	1.029	0.329	0.002
Fear of infection	1.399	0.237	<0.0001	1.477	0.174	<0.0001	2.876	0.374	<0.0001
Fear of social stigma	1.210	0.207	<0.0001	0.559	0.152	0.000	1.769	0.327	<0.0001
Disruption of daily life	0.017	0.007	0.021	0.007	0.005	0.206	0.024	0.011	0.04
Financial problem	0.806	0.172	<0.0001	0.231	0.126	0.068	1.037	0.271	0.000
Experience of COVID-19 (No, reference)									
Yes	1.667	0.395	<0.0001	1.052	0.289	0.000	2.719	0.622	<0.0001
Social support (No, reference)									
Yes	−0.817	0.567	0.151	−0.526	0.415	0.206	−1.340	0.893	0.134
Media consumption (Less than 1 h per day, reference)									
≥1 h per day	1.100	0.370	0.003	1.091	0.271	<0.0001	2.191	0.583	0.000

## Data Availability

The data presented in this study are available on request from the corresponding author.

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
