# Peer review of "Peritraumatic Distress during the COVID-19 Pandemic in Seoul, South Korea"

_ijerph, 2021, doi:10.3390/ijerph18094689_

Round 1
Reviewer 1 Report
In this paper, the authors evaluate the level of peritraumatic distress given the COVID-19 Pandemic. For this matter, they performed a questionnaire with 813 participants using an online platform. They used previously established formulae to calculate the peritraumatic distress level. In their results, the authors claim that the overall level of peritraumatic distress was slightly high when compared to literature values.
Overall, the text is well written. The authors performed a great job in describing the motivations for the work in the introductory section. Also, they thoroughly describe the process of data acquisition and individual parameters to evaluate. Finally, they present a broad evaluation of the results obtained from the data, exploring all values from the questionnaire.
There are two main concerns about this work. The first one is the lack of a systematical tool inside the text to evaluate the overall result. The authors claim that the overall value for the peritraumatic distress level is slightly high. Within the text, the only reference to support this argument is the comparison with the PTSD score threshold (23). The comparison with other countries peformed in the beginning of Section 4 is inconclusive in this regard. The text needs to establish systematically what is the criteria to classify the peritraumatic stress levels (as high, slightly high, average, slightly low, and low, for instance).
The other main concern is the need for approval in the ethics comittee in Korea. Usually, studies with humans require approval from the institution's ethics commitee. The authors declare that the participants informed consent, but how can the editorial board verify it? Did the authors submit their research to the ethics commitee? How about information disclosure? These questions need to be answered before considering the text for publication.
Reviewer 2 Report
Summary of the study
The present study aimed to investigate the peritraumatic reaction to the Covid-19 pandemic in the population of Seoul, the capital of South Korea, and the associated risk factors for peritraumatic distress, such as risk perception, fear, stigma and Covid-19-related experiences. Data were collected through an online survey, that was conducted at the end of the first wave of Covid-19 in South Korea.
The level of peritraumatic distress – measured through the Peritraumatic Distress Inventory (PDI) – was such that one third of participants were at risk of developing clinically elevated peritraumatic distress and potentially later developing post-traumatic stress disorder (PTSD). Moreover, risk perception, fear and stigma showed a significant association with elevated distress. Higher levels of distress were also associated with poor mental health, a certain amount of media consumption and disruption of daily life and financial difficulties. That said, the authors of this manuscript conclude by claiming that there is a need to develop evidence-based mental health interventions and to plan and adopt measures that can help recover to daily life.
Abstract
The present section summarizes in a very clear and extensive way all the main sections of this study.
Introduction
The present section presents in a very clear and appropriate way the theoretical background for the present study, providing readers with an overview of the international literature available on the discussed topics. The authors first describe, in general terms, the Covid-19 pandemic and Covid-19-related measures to later move on to the mental health consequences. Later, the authors underline a paucity of literature concerning trauma-related reactions and PTSD as a result of the pandemic. Moreover, they claim that just few research studies covered the situation in South Korea. At the end of the introduction, the authors very clearly state the objectives of the present study.
Although the introductory part is very clear and well-organized, it would be very helpful for readers if a definition of peritraumatic distress were to be provided and of the relationship between peritraumatic distress and PTSD. Moreover, the authors should provide appropriate references (scientific literature or news from newspapers) in support of the claims made between line 84 and 90 of page 2. Finally, as a minor consideration, the authors are invited to possibly consider substituting the expression “and so on” (page 1, line 50 and page 2 line 72) which does not seem appropriate within the context of a scientific article.
Materials & Methods
In this section the methodology is explained in such a detailed way as to allow a possible replication of this study. The following is a more in-depth description of each paragraph.
In the “Design and Participants” paragraph the sampling is explained in an extensive way. However, the authors might consider explaining how they obtained the e-mail addresses to which the survey was sent.
In the “Questionnaire” paragraph, the authors provide readers with an exhaustive description of all the measures and instruments employed to measure all the variables of interest in this study.
In the “Statistical Analysis” paragraph, the authors provide readers with a description of the statistical analysis made and with the criteria for statistical significance employed.
Results
In accordance with the journal guidelines, the results in this section are presented in a concise and precise way. The tables presented offer a clear and more immediate understanding of the data presented within the text.
Discussion
In the present section, the results emerged in the previous section are interpreted and considered within a broader context, by confronting them with other available data on the Covid-19 pandemic coming from other nations in the world. As suggested by the journal guidelines, the authors clearly state the limitations of the present study and possible future research. As a matter of fact, it would be interesting – as stated by the authors themselves – to run a longitudinal study as to capture possible fluctuations of the levels of peritraumatic distress in time connected to the variations in the severity of the measures adopted to combat the pandemic and, maybe, to the advent of the vaccine.
Finally, as a minor consideration, the authors should please provide readers with references in support of their claims at page 11, between lines 346 and 357.
Conclusions
This sections clearly presents the clinical and general implications of the results of the present study, outlining the need for the development of evidence-based and targeted public mental health intervention along with other measures that can help the population slowly go back to their daily lives.
References
This section is perfectly in line with the journal guidelines. The authors provide readers with an appropriate amount of updated international literature. The presented references are organized in a correct way and DOIs are always present when available. As a minor change, the authors should delete the comma between the authors of a paper and the title of the paper, as to fully conform with the journal guidelines.
Use of English
The English language used in this article is appropriate and it allows readers to fully comprehend the whole manuscript.
Overview
The present manuscript investigates a very relevant and very actual topic, that is Covid-19 and the risk factors and mental health consequences associated to it. One of the major assets of the present study is that it presents data based on a large sample. The description of all the steps made in this study are described in an optimal way, resulting in a very clear and efficient product. We therefore invite the authors to consider the hereby mentioned possible corrections which would make the article ready for publication.
Reviewer 3 Report
In relation to the Study's limitations, it will be necessary to include the sampling method and the bias in relation to the age of the population and their skills to access and use an electronic questionnaire.
Round 2
Reviewer 1 Report
I recommend the acceptance of this work, as the authors provided the necessary information.